# [^18^F]FES PET Resolves the Diagnostic Dilemma of COVID-19-Vaccine-Associated Hypermetabolic Lymphadenopathy in ER-Positive Breast Cancer

**DOI:** 10.3390/diagnostics13111851

**Published:** 2023-05-25

**Authors:** Munenobu Nogami, Tetsuya Tsujikawa, Hiroyuki Maeda, Nobuyuki Kosaka, Mizuho Takahashi, Naoki Kinoshita, Tetsuya Mori, Akira Makino, Yasushi Kiyono, Takamichi Murakami, Takanori Goi, Hidehiko Okazawa

**Affiliations:** 1Biomedical Imaging Research Center, University of Fukui, Fukui 910-1193, Japan; 2Department of Radiology, Kobe University Hospital, Kobe 650-0017, Japan; 3Department of Radiology, Faculty of Medical Sciences, University of Fukui, Fukui 910-1193, Japan; 4First Department of Surgery, Faculty of Medical Sciences, University of Fukui, Fukui 910-1193, Japan

**Keywords:** COVID-19, vaccination, lymphadenopathy, FES PET, breast cancer

## Abstract

Coronavirus disease (COVID-19) vaccination is known to cause a diagnostic dilemma due to false-positive findings on [^18^F]FDG PET in vaccine-associated hypermetabolic lymphadenopathy. We present two case reports of women with estrogen-receptor (ER)-positive cancer of the breast who were vaccinated for COVID-19 in the deltoid muscle. [^18^F]FDG positron emission tomography (PET) demonstrated primary breast cancer and multiple axillary lymph nodes with increased [^18^F]FDG uptake, diagnosed as vaccine-associated [^18^F]FDG-avid lymph nodes. Subsequent [^18^F]FES PET revealed single axillary lymph node metastasis in the vaccine-associated [^18^F]FDG-avid lymph nodes. To the best of our knowledge, this is the first study showing the usefulness of [^18^F]FES PET in diagnosing axillary lymph node metastasis in COVID-19-vaccinated patients harboring ER-positive breast cancer. Thus, [^18^F]FES PET has potential applications in the detection of true-positive metastatic lymph nodes in patients with ER-positive breast cancer regardless of the ipsilateral or contralateral side, who have received COVID-19 vaccination.

## 1. Introduction

Coronavirus disease (COVID-19) vaccination has caused a diagnostic dilemma due to false-positive findings on positron emission tomography (PET) with 2-[^18^F]fluoro-2-deoxy-D-glucose ([^18^F]FDG) in vaccine-associated hypermetabolic lymphadenopathy, particularly in patients with breast cancer [1,2]. The number of vaccine doses and the interval after last vaccination were found to be closely related to the presence of vaccine-associated hypermetabolic lymphadenopathy [2]. Accumulation in the axillary lymph nodes after vaccine administration has been reported in 14.5% to 53.9% of cases [3]. [^18^F]FDG uptake in the lymph nodes is more frequent in the case of Moderna compared to Pfizer-BioNTech vaccines [4]. Patients with a normal absolute lymphocyte count after COVID-19 vaccination were more likely to have [^18^F]FDG uptake in the lymph nodes [5]. The Society of Breast Imaging recommends obtaining information on the COVID-19 vaccination status, timing, and the side (left or right arm) of vaccination and scheduling breast imaging screening to take place 4 to 6 weeks after the second COVID-19 vaccination dose when possible. However, the incidence, timing, and characteristics of COVID-19-related hypermetabolic lymphadenopathy remain uncertain. PET with 16α-[^18^F]-fluoro-17β-estradiol ([^18^F]FES), an ^18^F-labelled compound of estradiol, is well-established for the diagnosis, staging, and post-therapeutic follow-up of patients with estrogen-receptor (ER)-positive breast cancer [6,7,8]. In 2020, the US Food and Drug Administration approved a PET imaging agent specifically for detecting ER-positive lesions as an adjunct to biopsy in patients with recurrent or metastatic breast cancer. Here, we report on two cases to evaluate the usefulness of [^18^F]FES PET in diagnosing axillary lymph node metastasis in COVID-19-vaccinated patients with ER-positive breast cancer. To the best of our knowledge, this is the first study to evaluate the usefulness of [^18^F]FES PET in diagnosing axillary lymph node metastasis in COVID-19-vaccinated patients with ER-positive breast cancer.

## 2. Case Presentation

The study was approved by the institutional review board (protocol code: 20108041, date of approval: 21 February 2017), and the requirement for informed patient consent was waived due to the retrospective nature of the study.

### 2.1. Case 1

A 59-year-old woman was diagnosed with ER-positive left breast cancer on histopathological examination, including hematoxylin and eosin (H&E) staining and immunohistochemistry (IHC) (ER 90%, PgR 30%, HER2 score1, MIB-1 index 30%) She underwent [^18^F]FDG PET/magnetic resonance imaging (MRI) for preoperative staging. The primary lesion in the breast, as well as multiple level I and II left axillary lymph nodes, showed significant FDG uptake (Figure 1a and Figure 2a–c). The patient had received the COVID-19 vaccine in the upper left arm 3 days prior to examination, and there was marginal FDG accumulation in the left deltoid muscle, suggesting inflammatory changes due to vaccine administration. Nineteen days after [^18^F]FDG PET/MRI, [^18^F]FES PET/MRI was performed again to assess ER expression in the lymph nodes (Figure 1b and Figure 2d,e). The largest axillary lymph node showed considerable FES uptake (Figure 1b and Figure 2d, red arrow), a finding consistent with lymph node metastasis; however, the other nodes with FDG accumulation did not show remarkable FES uptake (Figure 1b and Figure 2e,f, white arrows), indicating that the FDG uptake was due to inflammatory changes associated with vaccine administration. Uterine leiomyoma, another estrogen-dependent tumor, showed intense [^18^F]FES uptake in the pelvis (Figure 1b, yellow arrowhead).

The patient underwent left mastectomy and left axillary lymph node dissection. Histopathological diagnosis of the excised specimen of the left breast lesion via H&E staining led us to diagnose the lesion as invasive ductal carcinoma with intraductal extension (pT2, Figure 3a). Of the 20 dissected left axillary lymph nodes, only the largest excised lymph node specimen was diagnosed as lymph node metastasis via H&E staining (Figure 3b). The patient underwent postoperative chemotherapy and has not relapsed thus far.

### 2.2. Case 2

A 58-year-old woman was diagnosed with ER-positive right breast cancer on histopathological examination using H&E staining and IHC (ER 80%, PgR 80%, HER2 score1, MIB-1 index 10%). She underwent pretreatment [^18^F]FDG PET/CT for staging. In addition to intense [^18^F]FDG uptake in the primary breast tumor (Figure 4a and Figure 5a, red arrowhead), increased [^18^F]FDG uptake was observed in level I (Figure 4a and Figure 5b, red arrow) and level II contralateral axillary lymph nodes (Figure 4a and Figure 5c, white arrow). Since she was vaccinated against COVID-19 (the Moderna COVID-19 vaccine) in her left deltoid muscle 12 days before the [^18^F]FDG PET/CT scan, the [^18^F]FDG-avid contralateral axillary lymph nodes were considered as vaccine-associated hypermetabolic lymphadenopathy. She had undergone surgery for the right axillary ganglion, and we suspected possible altered lymphatic drainage. Four weeks after [^18^F]FDG PET/CT, she underwent [^18^F]FES PET/MRI to evaluate the ER expression of the lesions. [^18^F]FES PET/MRI showed moderate [^18^F]FES uptake in the ER-positive right breast cancer (Figure 4b and Figure 5d, red arrowhead) and high [^18^F]FES uptake in the level I contralateral axillary lymph node (Figure 4b and Figure 5e, red arrow), and her condition was finally diagnosed as synchronous contralateral axillary lymph node metastasis (CAM). In contrast, the level II lymph nodes did not show significant [^18^F]FES uptake (Figure 4b and Figure 5f, white arrow). Although histopathological evidence was not available, the lymph node with FES accumulation diminished after treatment with an aromatase inhibitor (Letrozole) and a molecularly targeted agent (Abemaciclib), indicating lymph node metastasis. Uterine leiomyoma with [^18^F]FES uptake was also observed in this patient (Figure 4b, yellow arrowhead). 

## 3. Discussion

Although [^18^F]FDG PET plays an important role in breast cancer staging [9], one of its limitations is non-specific tracer accumulation, including that in tissues undergoing inflammatory reactions. The tracer uptake of ipsilateral supraclavicular and axillary lymphadenopathy lasts for at least three weeks after COVID-19 vaccination [10] and can be observed after more than six weeks [11]. Compared to the lymph node metastasis of breast cancer, accumulation in the axillary lymph nodes after vaccine administration is reported to be more common in level 2 and 3 regions [12] and can be diagnosed using radiomics based on PET as well as CT features [13], but it is difficult to differentiate between the two based on FDG accumulation alone. In contrast, FES PET is highly specific for breast cancers expressing ERs and can differentiate them from inflammatory lymph nodes, which is difficult to achieve with [^18^F]FDG PET [14,15]. In the present study, histopathological evidence suggested that [^18^F]FES PET can differentiate inflammatory lymph node enlargement after COVID-19 vaccine administration from lymph node metastasis, a difference which is difficult to distinguish using [^18^F]FDG PET. In addition, we could accurately identify CAM, which is even more complex.

Since their approval at the end of 2020, COVID-19 vaccines have been widely distributed worldwide, and reports on FDG PET findings after vaccine administration have been published since the first half of 2021 [2,16,17,18]. Vaccine-induced accumulation in the ipsilateral axillary lymph nodes is known to be observed not only with FDG but also with [^11^C] Choline [19], [^18^F]Choline [20], [^18^F] prostate-specific membrane antigen (PSMA) [20], [^68^Ga]DOTA-peptide (DOTATATE) [21], and [^177^Lu] Lu-DOTATATE PRRT post-therapy planar scintigraphy, as well as single-photon emission computed tomography with computed tomography (SPECT/CT) [22]. On the other hand, [^68^Ga] fibroblast-activation protein inhibitor (FAPI) was reported to show no accumulation in the enlarged axillary lymph nodes after vaccine administration, similar to our report on [^18^F]FES, which may be useful in distinguishing lymph node metastasis [23].

CAM is a rare finding in patients with breast cancer, with an incidence rate between 1.9 and 6% [24]. This condition can be synchronous (found at the time of primary breast cancer diagnosis) or metachronous (found following prior breast cancer treatment as a recurrence). In the present case, the synchronous CAM could be caused by the alternative lymphatic drainage routes developed following the previous axillary surgery. Regardless of the CAM origin, breast cancer metastasis to CAM is considered as an M1, stage IV disease, and the detection of CAM is critical for determining the appropriate treatment [25,26]. Although synchronous CAM and vaccine-associated hypermetabolic lymphadenopathy existed concurrently in this patient with breast cancer, [^18^F]FDG PET/CT could not distinguish between them. In contrast, [^18^F]FES PET clearly delineated ER-positive synchronous CAM from vaccine-associated hypermetabolic lymphadenopathy.

This study has certain limitations: it is a case report of only two cases, and we cannot draw conclusions with a high level of clinical evidence. However, breast cancer cases in which COVID-19 vaccine was administered followed by both FDG-PET and FES-PET examinations are rare, and we believe that accumulating findings from these kinds of case reports will be useful for the diagnosis and treatment of breast cancer patients in the future.

## 4. Conclusions

In conclusion, [^18^F]FES PET combined with [^18^F]FDG characterized the distinct phenotypic features of the lesions. [^18^F]FES PET can be more specific than [^18^F]FDG for lymph node staging in patients with ER-positive breast cancer, independent of the cause of lymph node uptake.

## Figures and Tables

**Figure 1 diagnostics-13-01851-f001:**
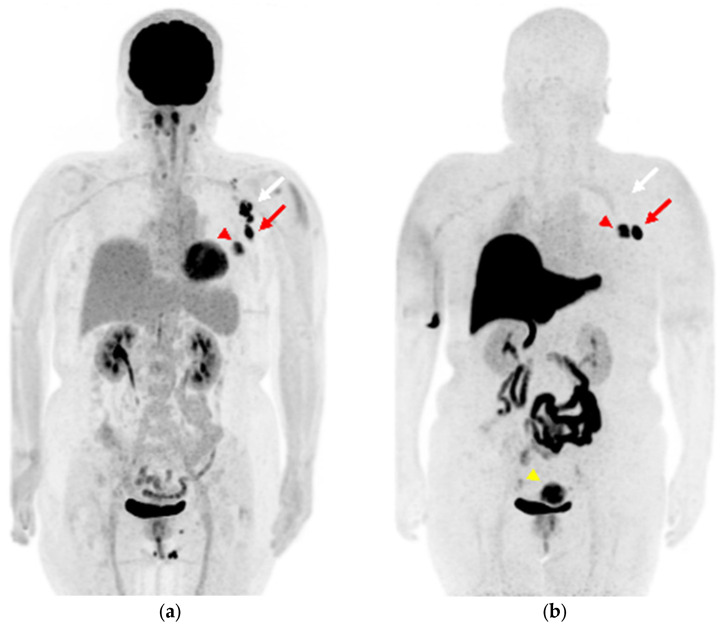
(**a**) Maximum-intensity projection (MIP) of [^18^F]FDG positron emission tomography (PET) image; (**b**) MIP image of [^18^F]FES PET. ((**a**), red arrowhead) The left breast cancer shows intense [^18^F]FDG uptake and ((**b**), red arrowhead) [^18^F]FES uptake. ((**a**), arrows) The ipsilateral axillary lymph nodes show intense [^18^F]FDG uptake, ((**b**), arrows) while [^18^F]FES PET is hyperintense only in the lesion with the red arrow. ((**b**), yellow arrowhead) Uterine leiomyoma shows intense [^18^F]FES uptake in the pelvis.

**Figure 2 diagnostics-13-01851-f002:**
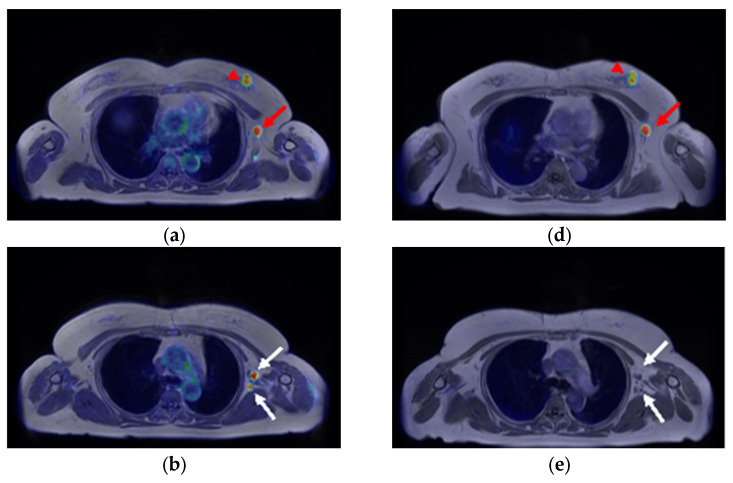
(**a**–**c**) [^18^F]FDG axial PET/magnetic-resonance (MR)-fused images and (**d**–**f**) [^18^F]FES PET/MR-fused images of left breast cancer and multiple ipsilateral axillary lymph nodes. The left breast cancer shows significant uptake on both ((**a**), arrowhead) [^18^F]FDG and ((**d**), arrowhead) [^18^F]FES PET. ((**a**–**c**), arrows) The multiple ipsilateral axillary lymph nodes show intense [^18^F]FDG uptake. ((**d**), red arrow) The largest ipsilateral axillary lymph node in level I shows intense [^18^F]FES uptake, whereas ((**e**,**f**), white arrows) the other nodes do not show significant uptake.

**Figure 3 diagnostics-13-01851-f003:**
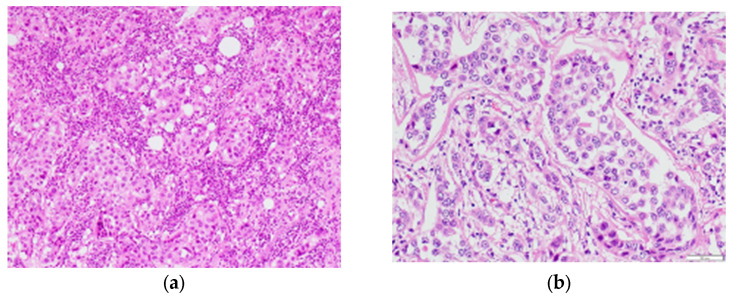
(**a**) High-power field image (×10) of a hematoxylin-eosin-stained specimen of the left breast lesion. Infiltrative growth of atypical cells with focal, ductal, and restiform structures is observed, with severe fibrosis and lymphocytic infiltration into the surrounding area. (**b**) High-power field image (×20) of a lymph node lesion specimen. Of the 20 resected lymph nodes, 1 showed evidence of cancer metastasis.

**Figure 4 diagnostics-13-01851-f004:**
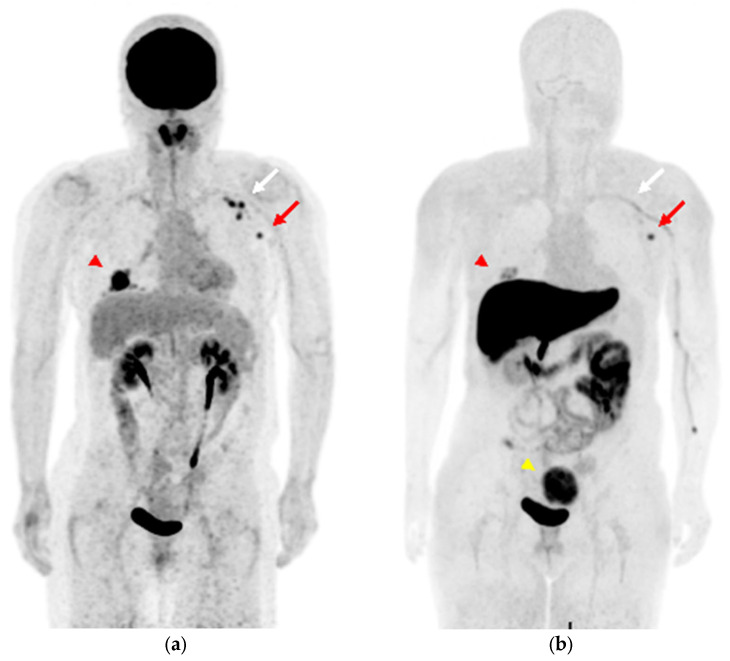
(**a**) MIP image of [^18^F]FDG PET, (**b**) MIP image of [^18^F]FES PET. ((**a**,**b**), red arrowhead) The right breast cancer sh6ws [^18^F]FDG and [^18^F]FES uptake. ((**a**,**b**), red arrows) Level I and ((**a**,**b**), white arrows) level II contralateral axillary lymph nodes show intense [^18^F]FDG uptake, while [^18^F]FES PET is hyperintense only in level I the lesion, shown with the red arrow. ((**b**), yellow arrowhead) Uterine leiomyoma shows intense [^18^F]FES uptake in the pelvis.

**Figure 5 diagnostics-13-01851-f005:**
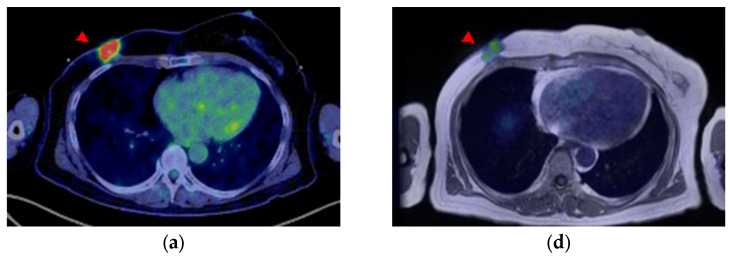
(**a**–**c**) [^18^F]FDG axial PET/computed-tomography (CT)-fused images and (**d**–**f**) [^18^F]FES PET/MR-fused images of right breast cancer and multiple contralateral axillary lymph nodes. ((**a**,**d**), arrowhead) The right breast cancer shows uptake on both [^18^F]FDG and [^18^F]FES PET. ((**a**–**c**), arrows) The multiple contralateral axillary lymph nodes show intense [^18^F]FDG uptake. ((**e**), red arrow) The level I contralateral axillary lymph node shows intense [^18^F]FES uptake, whereas ((**f**), white arrow) the other node does not show significant uptake.

## Data Availability

The minimal dataset is included within the manuscript. Additional data used for analysis are available from the corresponding author. The clinical data and images, including personally identifiable information, were not permitted to be disclosed by the ethical committee in our institution (Research Ethics Committee of the University of Fukui) and are not accessible due to laws on the protection of personal information in our country. Please direct further data inquiries to the Research Ethics Committee of the University of Fukui.

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
