# Peer review of "[18F]FES PET Resolves the Diagnostic Dilemma of COVID-19-Vaccine-Associated Hypermetabolic Lymphadenopathy in ER-Positive Breast Cancer"

_diagnostics, 2023, doi:10.3390/diagnostics13111851_

Round 1
Reviewer 1 Report
The manuscript intends to present two cases of women with breast cancer who underwent PET/CT with both FDG and FES soon after COVID-19 vaccination, who showed axillary lymph nodes with diverse tracer uptake.
The topic is certainly of interest to the readers, as false-positive findings are a tricky issue in such patients; at present time, the "solution" provided is not useful anymore as, fortunately, in recent months COVID-19 pandemic has fallen down significantly. For this reason, the sentence "can be used for staging in COVID-19-vaccinated patients with ER-positive breast cancer" in the conclusion has less sense now, as the number of vaccinations is decreasing and will be close to zero in a few months. A different conclusion should be written, for example highlighting the fact that FES is a more specific tracer than FDG for the detection of metastasis, independently from the cause of lymph node uptake (e.g. COVID-19 vaccination, granulomatous disease, etc.).
Furthermore, all the figures should be better presented; in particular, from the description I guess that you mix both tracers in a single figure (as you describes nodes with different uptake), but you should clarify better the layout of those images and the tracer administered.
Moreover, false-positive lymph node uptake has been described also with [177Lu] Lu-DOTA-TATE therapy, see Kolade, O.U., Ayeni, A.O., Brink, A. et al. SARS-CoV-2 vaccination site as possible pitfall on somatostatin receptor imaging. Clin Transl Imaging 10, 579–585 (2022). Please add this reference to the list.
English is fine and only minor editing is required.
Author Response
We thank the reviewer for evaluating of our study, and the editor for giving us the opportunity to resubmit our revised manuscript. We are happy to address all comments in a point-by-point manner. The revisions have been marked in a colored font for ease of identification in the revised version of our manuscript.
Comments: The topic is certainly of interest to the readers, as false-positive findings are a tricky issue in such patients; at present time, the "solution" provided is not useful anymore as, fortunately, in recent months COVID-19 pandemic has fallen down significantly. For this reason, the sentence "can be used for staging in COVID-19-vaccinated patients with ER-positive breast cancer" in the conclusion has less sense now, as the number of vaccinations is decreasing and will be close to zero in a few months. A different conclusion should be written, for example highlighting the fact that FES is a more specific tracer than FDG for the detection of metastasis, independently from the cause of lymph node uptake (e.g. COVID-19 vaccination, granulomatous disease, etc.).
Response: Thank you very much for your valuable suggestions. As you mentioned, COVID-19 is already converging and the frequency of encountering cases after vaccine administration is now decreasing. As you suggested, I have revised the description in the conclusion section (Page 8, line 165-167).
Comments: Furthermore, all the figures should be better presented; in particular, from the description I guess that you mix both tracers in a single figure (as you describes nodes with different uptake), but you should clarify better the layout of those images and the tracer administered.
Response: Thank you very much for your valuable feedback. As you mentioned, the figure explanation compares two tracers, while the figure shows only one tracer, which we assume causes some confusion. We have revised the figure to be divided into MIP and Fusion images, and to show the two tracers in comparison in each figure.
Comments: Moreover, false-positive lymph node uptake has been described also with [177Lu] Lu-DOTA-TATE therapy, see Kolade, O.U., Ayeni, A.O., Brink, A. et al. SARS-CoV-2 vaccination site as possible pitfall on somatostatin receptor imaging. Clin Transl Imaging 10, 579–585 (2022). Please add this reference to the list.
Response: Thank you for your valuable information. We have added the paper you mentioned to the references.
Reviewer 2 Report
The article “[18F]FES PET resolves the diagnostic dilemma of COVID-19 2 vaccine-associated hypermetabolic lymphadenopathy in ER-3 positive breast cancer” describes about usefulness of [18F]FES PET in diagnosing axillary lymph node metastasis in COVID-19-vaccinated patients having ER +ve breast cancer in case studies.
Following concerns need to be fixed in the ms.
1. Please mention the ethical approval number in section 2.
2. Two case studies are not enough to consider the results as a method rather it can be a diagnostic report only.
3. Materials methods for histo-chemical study must be presented.
The language quality of the article is ok.
Author Response
We thank the reviewer for evaluating of our study, and the editor for giving us the opportunity to resubmit our revised manuscript. We are happy to address all comments in a point-by-point manner. The revisions have been marked in a colored font for ease of identification in the revised version of our manuscript.
Comments: Please mention the ethical approval number in section 2.
Response: Thank you for pointing this out. We have added the Ethics Committee approval number to section 2.
Comments: Two case studies are not enough to consider the results as a method rather it can be a diagnostic report only.
Response: Thank you for your valuable comments. As you mentioned, this study has certain limitations: it is a case report of only two cases, and we cannot draw any conclusions with a high level of medical evidence. However, breast cancer cases in which COVID-19 vaccine was administered and both FDG-PET and FES-PET examinations were performed are rare, and we believe that the accumulation of knowledge from this type of case report will be useful for the diagnosis and treatment of future breast cancer patients. We have added the above statement to the end of the Discussion section.
Comments: Materials methods for histo-chemical study must be presented.
Response: Thank you for your important remarks. I have added the histopathological examination method of the primary lesion in each case (Page 2, line 58-60; page 5, line 102, 103), respectively. The histopathological examination procedure for the lymph node lesion in case 1 is described in the figure legend (Figure 3).
Round 2
Reviewer 1 Report
The manuscript has been extensively improved now, but still the captions of Figures 2 and 5 need to be checked and corrected (e.g. in Fig5 you say LEFT cancer, but it is RIGHT; in Fig2 please better separate arrows and arrowheads in the description).
Minor editing of English language required
Author Response
We thank again the reviewer for evaluating our study. We are happy to address all comments in a point-by-point manner. In addition, the entire text was proofread for English (www.editage.com) once again. The revisions have been marked in a colored font for ease of identification in the revised version of our manuscript.
Comments: The manuscript has been extensively improved now, but still the captions of Figures 2 and 5 need to be checked and corrected (e.g. in Fig5 you say LEFT cancer, but it is RIGHT; in Fig2 please better separate arrows and arrowheads in the description).
Response: Thank you for pointing out this very important point. The left and right errors have been corrected, and the arrows and arrowheads in Figure 2 have been described separately (Figure 2, line 85-87; figure4, line 134). However, the explanations for arrows b and c or e and f are the same, so they are shown together.
Reviewer 2 Report
It can be accepted now provided that the materials methods must be provided with histochemical studies procedures not in the figure legend.
ok
Author Response
We thank again the reviewer for evaluating our study. We are happy to address all comments in a point-by-point manner. In addition, the entire text was proofread for English (www.editage.com) once again. The revisions have been marked in a colored font for ease of identification in the revised version of our manuscript.
Comments: It can be accepted now provided that the materials methods must be provided with histochemical studies procedures not in the figure legend.
Response: Thank you for your comments. According to your suggestions, methods for histopathological examinations were described not only in the figure legends but also in the main body of the manuscript (Page 4, line 90-95) .